# Supplementation with High or Low Iron Reduces Colitis Severity in an AOM/DSS Mouse Model

**DOI:** 10.3390/nu14102033

**Published:** 2022-05-12

**Authors:** Seonghwan Moon, Minju Kim, Yeonhee Kim, Seungmin Lee

**Affiliations:** Department of Food and Nutrition, BK21 FOUR Project, College of Human Ecology, Yonsei University, Seoul 03722, Korea; moo15@yonsei.ac.kr (S.M.); minjue3517@yonsei.ac.kr (M.K.); yh0227@yonsei.ac.kr (Y.K.)

**Keywords:** iron deficiency, iron overload, colitis-associated colorectal cancer, cancer iron metabolism, inflammatory bowel disease, AOM/DSS mouse model

## Abstract

The relationship between colitis-associated colorectal cancer (CAC) and the dysregulation of iron metabolism has been implicated. However, studies on the influence of dietary iron deficiency on the incidence of CAC are limited. This study investigated the effects of dietary iron deficiency and dietary non-heme iron on CAC development in an azoxymethane/dextran sodium sulfate (AOM/DSS) mouse model. The four-week-old mice were divided into the following groups: iron control (IC; 35 ppm iron/kg) + normal (NOR), IC + AOM/DSS, iron deficient (ID; <5 ppm iron/kg diet) + AOM/DSS, and iron overload (IOL; approximately 2000 ppm iron/kg) + AOM/DSS. The mice were fed the respective diets for 13 weeks, and the AOM/DSS model was established at week five. FTH1 expression increased in the mice’s colons in the IC + AOM/DSS group compared with that observed in the ID and IOL + AOM/DSS groups. The reduced number of colonic tumors in the ID + AOM/DSS and IOL + AOM/DSS groups was accompanied by the downregulated expression of cell proliferation regulators (PCNA, cyclin D1, and c-Myc). Iron overload inhibited the increase in the expression of NF-κB and its downstream inflammatory cytokines (IL-6, TNFα, iNOS, COX2, and IL-1β), likely due to the elevated expression of antioxidant genes (*SOD1*, *TXN*, *GPX1*, *GPX4*, *CAT*, *HMOX1*, and *NQO1*). ID + AOM/DSS may hinder tumor development in the AOM/DSS model by inhibiting the PI3K/AKT pathway by increasing the expression of *Ndrg1*. Our study suggests that ID and IOL diets suppress AOM/DSS-induced tumors and that long-term iron deficiency or overload may negate CAC progression.

## 1. Introduction

Colorectal cancer (CRC) is the third most common cancer and has the second-highest mortality rate worldwide [1]. Despite recent developments and improvements in various treatment strategies, the mortality rate of CRC remains high [2]. Inflammatory bowel disease (IBD) has been suggested as one of the strong risk factors for CRC. IBD may increase the estimated risk of CRC development from 1 to 18% to a higher range depending on age, severity, duration of the disease, and degree of inflammation in colitis or polyps [3,4]. IBD is a chronic inflammatory condition of the gastrointestinal tract that includes ulcerative colitis (UC) and Crohn’s disease (CD) [5]. Chronic inflammation can induce tumorigenesis by increasing oxidative stress and epithelial cell proliferation and promoting angiogenesis [6,7]. In addition, recurrent inflammation is associated with tumors through mechanisms, such as the generation of mucosal mediators and changes in immune receptor expression in epithelial cells [8].

Iron is an indispensable nutrient for cellular metabolism and is involved in oxidative metabolism, cell proliferation, inflammation, and various in vivo reactions [9]. Iron homeostasis is tightly regulated in normal cells. For example, high iron levels induce the inactivation or decomposition of iron regulatory proteins (IRPs), thereby increasing the translation of the iron exporter ferroportin (FPN) and iron storage protein subunit ferritin heavy chain (FTH), while the degradation of FPN is inhibited by reduced hepcidin expression [10,11,12]. Contrastingly, in ID environments, hypoxia-inducible factor 1-alpha (HIF1-α) and IRP are activated to increase the mRNA level of iron importer transferrin receptor 1 (*TfR1*) and suppress the translation of FTH1 and FPN [13].

Generally, compared to normal colon cells, CRC cells are characterized by high dependence on iron, showing high iron requirements and elevated iron import [14,15]. Some human studies have revealed that increased iron intake is associated with a higher risk of IBD and CRC compared to adequate iron intake [16,17,18,19,20,21]. In a case-control study in Japan, Kobayashi et al. reported an increased odds ratio for UC (OR = 4.05 (1.46–11.2)) in the group with the highest dietary iron intake (over 7.18 mg/2000 kcal) compared to that with the lowest intake (under 5.87 mg/2000 kcal) [18]. Furthermore, in a European case-control study, circulating serum iron (OR = 1.17 (1.00–1.36)) was significantly associated with the risk of CRC (*p* = 0.049) [22]. However, other human studies have reported no or marginally significant associations between dietary iron intake and IBD or CRC [21,22,23]. In a prospective cohort study using dietary information of 165,311 women in the US, researchers found no significant association between dietary heme iron or total iron intake and the risk of CD or UC [21]. In another cohort study conducted in Canada, the risk of CRC was not associated with the intake of iron, heme iron, or iron from meat [23]. In addition, in a recent animal study using the DSS model, short-term exposure (8–10 days) to a low- or high-iron diet (100 ppm (equal to 20 mg Fe/kg diet) or 400 ppm ferrous iron, respectively) induced an approximately three times higher colitis score than that of normal chow (200 ppm ferrous iron). In contrast, long-term (8 weeks) exposure showed no significant influence on the score [24]. Although several studies have been published on the association between iron and colon tumorigenesis [25,26,27], the results are still controversial [28].

IBD patients suffer from oxidative stress caused by continuously generated reactive oxygen species (ROS) and recurrent inflammatory reactions through increased expression of inflammatory cytokines [29]. During an immune response, excessive ROS induces oxidative stress, and when it persists, can in turn can cause chronic inflammation [30]. Thus, ROS has been proposed as a causal and permanent factor of IBD and has been suggested to act as a regulatory factor of several inflammatory cytokines [31]. Iron may play a role in exacerbating colitis because of its ability to induce ROS production through a Fenton reaction [32]. Carrier et al. reported that in colitis-induced mice, iron supplementation (3%/kg diet carbonyl iron) caused increased lipid peroxidation levels and inflammatory responses by an intracellular Fenton reaction compared to the control (0.027%/kg diet carbonyl iron) [33]. Changes in iron metabolism have been reported, along with an increase in iron content in tumors [34]. An elevated expression of hepcidin in the liver and TfR1 and divalent metal transporter 1 (DMT1) in the colon was observed in a colitis mouse model [35,36]. Hepcidin expression increases due to the activation of inflammatory pathways, such as interleukin (IL)-6/signal transducer and activator of transcription 3 (STAT3) in IBD [37]. Iron depletion or supplementation may also affect the development of inflammation and cancer. In a DSS mouse model, iron depletion using iron chelators or an ID diet mitigated colitis [32,38]. Conversely, Anita et al. reported that high dietary iron (1% carbonyl iron) increased the expression of inflammatory cytokine IL-6 and STAT3 phosphorylation and worsened colitis and colitis-associated colorectal cancer (CAC) in azoxymethane/dextran sodium sulfate (AOM/DSS) mouse models [39]. IL-6 upregulates the phosphatidylinositol 3-kinase (PI3K)- protein kinase B (AKT) pathway, one of the key pathways commonly activated during tumor development in both CAC and CRC [40,41,42]. In contrast, Xu et al. suggested that an increase in iron-induced ROS in IBD promotes the development of CAC but simultaneously induces ferroptosis [43].

Since iron acts as a double-edged sword, which could instigate tumorigenesis or cell death [44], it is important to explore the effects of iron overload or deficiency on CAC. In addition, knowledge of the mechanistic role of iron depending on its intake level could lead to potential applications in the prevention or treatment of related diseases. In this study, we aimed to investigate the effects of dietary-induced overload or deficiency of iron on tumor development and changes in iron metabolism using AOM/DSS-induced CAC rodent models.

## 2. Materials and Methods

### 2.1. Animal Care and Experimental Protocol

The composition of the different iron diets is outlined in Table 1. Male C57BL/6J mice (3 weeks old) were purchased from Doo Yeol Biotech (Seoul, Korea). Animals were housed under controlled conditions of humidity (55 ± 10%), lighting (12 h light/dark cycle), and temperature (23 ± 2 °C) with access to pure water and pelleted basal diet. After one week of acclimation, the animals were randomly divided into the following groups (*n* = 29): IC + NOR (*n* = 5) and IC + AOM/DSS (*n* = 8), ID + AOM/DSS (*n* = 8), and IOL + AOM/DSS (*n* = 8) groups. The mice in the respective groups were fed an iron control (IC) diet (35 ppm iron/kg), iron-deficient (ID) diet (<5 ppm iron/kg diet), and iron overload (IOL) diet (about 2000 ppm iron/kg diet). Four weeks later, an AOM/DSS mouse model was developed. A single dose of AOM (10 mg/kg) was intraperitoneally injected in the IC + AOM/DSS, ID + AOM/DSS, and IOL + AOM/DSS groups. At week 7, the AOM groups were provided with 1% DSS (wt/vol) in drinking water for seven consecutive days, followed by 14 days of regular drinking water. This cycle was repeated twice. Body weights were measured once per week throughout the experiment. The same procedure was performed with normal saline and distilled drinking water for the IC + NOR group. In this process, two mice in the ID + AOM/DSS group died. The remaining mice were euthanized at week 13. All experimental protocols were approved by the Institutional Animal Care and Use Committee (IACUC) of Yonsei University, South Korea (permit number: 201711-660-03).

### 2.2. Morphological Analysis

The colon was excised from the mice in each group, and the number of tumors was counted. Furthermore, the ratio of tumors was calculated as follows: (tumor number)/(tumor number of AOM/DSS group) × 100.

### 2.3. Histological Analysis

To perform a histological analysis of the tissue samples, 10% buffered formalin-fixed colon tissues were dehydrated and embedded in paraffin blocks. After sectioning, the slices were deparaffinized and rehydrated using a xylene-ethanol-water gradient system. The sections were stained with hematoxylin and eosin (H&E) to confirm iron accumulation. For the morphological analysis using the Perls’ Prussian blue staining, sliced samples were incubated in a working solution of 1:1 20% HCl and 10% potassium ferrocyanide. After washing with distilled water, the slices were stained with a nuclear fast red solution. All representative photomicrographs of the samples were obtained using an Eclipse Ti light microscope (Nikon, Tokyo, Japan).

### 2.4. Inductively Coupled Plasma-Optical Emission Spectrometry (ICP-OES) and Inductively Coupled Plasma Mass Spectrometry (ICP-MS) Analysis

The iron content of the mouse liver was measured using ICP-OES (Optima 8300, Perkin Elmer, MA, USA) and ICP-MS (NexlOn300, Perkin Elmer, Waltham, MA, USA). Approximately 50 mg of freeze-dried tissues was dissolved in 8 mL of 60% HNO_3_ solution (Showa Chemical Industry, Tokyo, Japan) and 2 mL of 30% H_2_O_2_ solution (Junsei Chemical Co., Tokyo, Japan) and then adjusted to 50 mL using 1% HNO_3_ solution. Subsequently, it was loaded into a microwave digestion system (TOPEX, PreeKem, Shanghai, China) and pretreated according to the manufacturer’s instructions. A quality control standard (cat. IV-28, Inorganic Ventures, Christiansburg, VA, USA) and multi-element calibration standard 3 (N9300233, PerkinElmer Pure Plus, Waltham, MA, USA) were used. The iron concentration in the raw materials was calculated by multiplying the concentration measured after pretreatment with the dilution multiple of the pretreatment. Samples with a concentration of less than 0.1 ppm, which is less reliable during ICP-OES analysis, were also evaluated via ICP-MS.

### 2.5. cDNA Extraction and Reverse Transcription (RT)-PCR

Colon tissue samples were frozen in liquid nitrogen and mechanically dissociated in an RNA buffer. Total RNA was extracted using TRIzol reagent (Molecular Research Center, Inc., Cincinnati, OH, USA), according to the manufacturer’s protocol. RT-PCR was performed with 1 µg of total RNA to synthesize cDNA using a random primer mixture, according to the manufacturer’s instructions.

### 2.6. Western Blot

Protein from colon tissues was extracted with a standard radioimmunoprecipitation assay (RIPA) buffer supplemented with a cocktail containing 1 mM sodium diphosphate decahydrate, 1 mM β-glycerophosphate, 1 mM NaF, 1 mM Na_3_VO_4_, protease inhibitors, and phosphatase inhibitors. Protein samples were loaded in equal amounts onto 10% SDS-polyacrylamide gels and then separated. The samples were then transferred to PVDF membranes (IPVH00010, Merck Millipore, Billerica, MA, USA). The membrane was incubated with the primary antibodies (1:1000; *v*/*v*), such as mouse anti-FTH1, mouse anti-PCNA, rabbit anti-CyclinD1, mouse anti-NF-κB p65, mouse anti-c-Myc, rabbit anti-Bax, mouse anti-Bcl-2, mouse anti-IL-6, mouse anti-iNOS, mouse anti-TNFα, mouse anti-PTEN from Santa Cruz Biotechnology Inc. (Dallas, TX, USA), as well as rabbit anti-TfR1 and rabbit anti-pAKT from Cell Signaling Technology (Danvers, MA, USA). Overnight and horseradish peroxide (HRP)-conjugated secondary antibodies (1:5000; Bio-Rad Laboratories, Hercules, CA, USA) were used to detect the proteins. Membranes were washed with phosphate-buffered saline, and protein bands were visualized using a D-Plus^TM^ ECL Femto System (Dongin Biotech, Seoul, Korea). The blots were photographed using an AE-9300 Ez-Capture system (ATTO, Tokyo, Japan), and the levels of protein expression were quantified using ImageJ software (National Institutes of Health, Bethesda, MD, USA) and normalized using the internal standards.

### 2.7. Statistical Analysis

SPSS software was used to perform statistical analyses (IBM SPSS Statistics package version 25, IBM Corp., Armonk, NY, USA). All the results are presented as mean ± standard deviation (SD), except the mRNA data, which are presented as mean ± standard error (SE). Statistical significance was tested using one-way analysis of variance with Tukey’s post-hoc test. Student’s *t*-test was used for comparisons among experimental groups. A *p*-value < 0.05 was set as the criterion for statistical significance.

## 3. Results

### 3.1. Effects of ID and IOL Diets on Colon Length, Body Weight, and Body Iron Levels in the AOM/DSS Model

The timeline of the animal study is depicted in Figure 1a. When comparing the indicators that reflect the severity of colitis between the groups, we observed that the AOM/DSS-treated groups had shorter colon lengths than the IC + NOR group (Figure 1b). In particular, the ID + AOM/DSS group showed a significantly greater decrease in colon length, while no difference in colon length was found in the IOL + AOM/DSS group when compared with that in the IC + AOM/DSS group (Figure 1b). In addition, the weights of the mice just before euthanasia were significantly lower in the AOM/DSS groups than in the IC + NOR group (Figure 1c). In particular, the IC + AOM/DSS group showed a weight loss of approximately 18.9% compared with the IC + NOR group, even though there was no significant difference in the cumulative food intake between the IC + NOR group and the IC + AOM/DSS group (Figure 1c,d). This implied that the AOM/DSS mouse model was successfully established. The weights of the organs relative to body weight are shown in Appendix A. Overall, our data imply that iron insufficiency might be highly correlated with worsened colitis than excess iron, as the shortening of colon length and a decreased weight signify the aggravation of colitis.

To measure the differences in colon iron content between the groups, we used Perl’s Prussian blue staining. AOM/DSS treatment significantly reduced the iron level in the colon; however, the IOL diet group exhibited significantly increased colon iron content compared with the IC + NOR group (Figure 1e). The iron content of the liver was measured through ICP-OCE & ICP-MS. Similar to that of the large intestine, the iron content of the liver decreased after AOM/DSS treatment, but was significantly higher in the IOL + AOM/DSS group than in the IC + NOR group (Figure 1f).

### 3.2. Iron Metabolic Gene Expression Was Altered in the AOM/DSS-Induced Cancer Model and Further by the ID and IOL Diets

As we confirmed that the iron content in the liver and colon tissues was affected by AOM/DSS treatment or IOL diet, we examined the changes in the expression of iron metabolism genes and regulatory mechanisms in the AOM/DSS mouse model. The liver maintains systemic iron homeostasis and responds to excess iron by increasing hepcidin expression and storing excess iron in the liver using FTH1 [45]. The expression of liver hepcidin, a systemic regulator of iron metabolism [46], increased significantly in the IOL + AOM/DSS group compared to that in the other groups and decreased in the ID + AOM/DSS group compared to that in the IC + AOM/DSS group (Figure 2a). The expression of TfR1, an iron importer, increased only in the ID + AOM/DSS group liver compared with the others [47] (Figure 2b). Similar to those in the liver, mRNA and protein levels of TfR1 in the large intestine were significantly increased only by the ID diet among the AOM/DSS-treated groups (Figure 2e), suggesting that iron import may be induced to improve the low-iron status in both the liver and the large intestine (Figure 2b,e). The protein level of FTH1 in the liver was higher in the IC + AOM/DSS group than in the IC + NOR group and did not change among the AOM/DSS groups, although the ID + AOM/DSS group showed a marginal reduction without statistical significance (Figure 2c). The protein level of FTH1 in the colon was higher in the IC + AOM/DSS group than in the other groups, whereas, unlike those in the liver, FTH1 levels were significantly reduced in the ID + AOM/DSS and IOL + AOM/DSS groups (Figure 2d). The mRNA levels of DMT1 and FPN did not differ between the IC + NOR and IC + AOM/DSS groups, but were significantly upregulated in the IOL + AOM/DSS group compared with those in the other groups (Figure 2f,g). Overall, the data suggest that AOM/DSS caused the upregulation of FTH1, which was significant in the large intestine, but not in the liver, by either ID or IOL diets. However, colonic FTH1 protein levels were reduced by the ID and IOL diets. IOL diets induced the unexpected upregulation of DMT1 in AOM/DSS, but other iron regulatory genes, including hepcidin, TfR1, and FPN, were regulated by iron diets, as expected.

### 3.3. IOL and ID Diets Alleviated the Development of Colonic Tumors

To investigate the effects of high or low dietary iron on the development of colon cancer, the AOM/DSS groups were compared for morphological changes in the colon tissues by H&E staining, the number of tumors, and the expression levels of proliferation marker genes. All H&E-stained images are shown in Appendix A. Compared with the IC + NOR group, the IC + AOM/DSS group showed severe morphological changes (Figure 3a). Tumors were also formed in the colon tissue (Figure 3b). In addition, the total tumor area for each colon was defined as the tumor burden index [48] (Appendix A).

In addition, protein levels of proliferating cell nuclear antigen (PCNA), cyclin D1, and c-Myc were significantly higher in the IC + AOM/DSS group than in the IC + NOR group (Figure 3c–e). However, protein levels of PCNA and c-Myc were considerably lower in the ID + AOM/DSS and IOL + AOM/DSS groups than in the IC + AOM/DSS (Figure 3c–e). Only IOL + AOM/DSS exhibited a significant reduction in cyclin D1 expression (Figure 3d). As for Bcl-2-associated X protein (Bax)/B-cell lymphoma (Bcl)-2 ratio, which has been suggested to determine the susceptibility to apoptosis [49], there were statistical differences when comparing IC + NOR with IC + AOM/DSS and ID + AOM/DSS groups, while IOL + AOM/DSS showed a significantly higher ratio than those of all other groups (Figure 3f). The data suggest that both ID and IOL diets prior to AOM/DSS-induced colorectal cancer formation might alleviate the incidence of colorectal cancer by decreasing cellular proliferation by both diets and increasing apoptosis only by IOL.

### 3.4. IOL Diet Downregulated AOM/DSS-Induced Inflammatory Gene Expression

To explore the effects of high or low dietary iron on AOM/DSS-induced inflammation in the colon, the expression levels of pro-inflammatory genes related to the pathological pathways of CAC [50,51] and genes related to angiogenesis and tumor development [52] were analyzed. First, AOM/DSS increased the protein expression level of nuclear factor kappa-light-chain-enhancer of activated B cells (NF-κB) p65 (Figure 4a), a central mediator in the induction of inflammatory damage [53] and subsequently increased the expression levels of NF-κB target genes [54], including IL-6, tumor necrosis factor (TNF) α, inducible nitrous oxide (iNOS), cyclooxygenase 2 (COX2), and IL-1β (Figure 4b–f). These changes observed in the AOM/DSS group were reversed in the IOL + AOM/DSS group, but not in the ID + AOM/DSS group (Figure 4). The IOL + AOM/DSS group showed lower expression levels of NF-κB p65, IL-6, TNFα, iNOS, COX-2, and IL-1β than the IC + AOM/DSS group, whereas the ID + AOM/DSS group showed no changes in p65, IL-6, TNFα, and iNOS levels, and higher increases in COX-2 and IL-1β levels (Figure 4). Our findings suggest that IOL diet-induced iron overload might hinder AOM/DSS-induced inflammation, not observed in ID-induced iron deficiency.

### 3.5. Iron Overload Diet Enhanced Antioxidant System through Upregulating the Antioxidant Signaling Pathway

As our data demonstrated that dietary IOL was associated with a reduced number of tumors and AOM/DSS-induced inflammation, we attempted to reveal the underlying mechanism of the anti-inflammatory role of IOL in AOM/DSS-generated tumors. Oxidative stress and the inflammatory response caused by IBD may be suppressed by antioxidant enzymes, accompanied by a decrease in the expression of inflammatory cytokines [55]. The knockout of Nrf2, a representative antioxidant gene transcription factor, can increase susceptibility to AOM/DSS-induced colitis and tumorigenesis [56]. We assessed the expression levels of the antioxidant genes (Figure 5). The results showed that IOL + AOM/DSS mice had a higher expression of antioxidant genes known to be downstream genes of Nrf2. Specifically, IOL + AOM/DSS exhibited significant elevations in the gene expression of thioredoxin (*TXN*), glutathione peroxidase 1 (*GPX1*), catalase (*CAT*), heme oxygenase 1 (*HMOX1*), and NAD(P)H quinone dehydrogenase 1 (*NQO1*), which showed lower expression because of AOM/DSS treatment (Figure 5b,c,e–g). The collective data suggested the possible effects of increased expression of antioxidative genes on IOL-inhibited inflammation in CAC.

### 3.6. Iron-Deficient and -Overload Status with AOM/DSS Affected Tumor Suppression through PI3K/AKT Pathway

To explore other possible mechanisms by which dietary ID or IOL affects tumor volume reduction, we specifically assessed PI3K/AKT pathway genes that interfere with cell proliferation and survival in various cancers, including CRC [57]. First, the level of phosphorylated AKT (pAKT), an activated form of AKT, was higher in IC + AOM/DSS than in IC + NOR, whereas other AOM/DSS-treated groups showed no statistically significant differences (Figure 6a). Phosphatase and tensin homolog (PTEN), a negative regulator of the PI3K/AKT pathway [57], were upregulated significantly in ID + AOM/DSS mice and marginally in IOL + AOM/DSS mice, compared with those in IC + AOM/DSS mice (Figure 6b). In addition, the mRNA levels of N-Myc downstream regulated 1 (*Ndrg1*), a tumor metastasis suppressor [58], were elevated in the ID + AOM/DSS group, but not in the IOL + AOM/DSS group, compared with those in the IC + AOM/DSS group (Figure 6c). These data might suggest that both dietary ID and IOL hindered the proliferation of AOM/DSS-generated tumors by suppressing the PI3K/AKT pathway. ID diet induced this in part through an increased expression of *Ndrg1*, which was not observed in IOL diet-fed mice.

## 4. Discussion

In this study, we report the alleviating effects of disturbed iron status through long-term high- or low-iron diets on colon cancer development in an AOM/DSS mouse model, which might be associated with an altered *FTH* gene expression and diminished inflammation and cell proliferation. Although several studies have reported significant changes in the incidence of cancer development by adjusting dietary iron intake [25,39], there is a lack of studies comparing iron metabolism and status in AOM/DSS-treated mice by adjusting iron doses—iron overload or deficiency.

First, we observed aberrant iron status and metabolism in AOM/DSS-induced CRC mice and compared the effects of iron overload or deficiency on the development of colon cancer. In previous clinical trials of CRC patients, iron accumulation was higher in tumor tissues than in normal tissues [59,60]. Although we could not measure iron content specifically in the tumor mass, there was a significant decrease in total iron content in the liver and colon tissues of AOM/DSS-treated mice, suggesting that body iron levels might be diminished during colon cancer development. In contrast to our results, Kim et al. reported no significant differences in the iron content of the liver between the control group and AOM/DSS [61]. The difference between these results could be because of the differences in the duration of iron diet administration. Our diet period (13 weeks) was shorter than that of Kim et al. (24 weeks). In addition, despite the lower iron levels in the AOM/DSS mice observed in our study, colonic expression of FTH1 was elevated in AOM/DSS mice compared to that in normal mice. High ferritin expression levels in cancer cells have been suggested to lower the cytoplasmic LIP levels [62], which might increase resistance to oxidative stress and DNA oxidative damage, inhibit apoptosis [63], and increase angiogenesis and tumor progression through the upregulation of VEGF [64]. In addition, Svitlana et al. reported that an increase in FTH1 expression might augment the possibility of malignant breast cancer progressing to aggressive tumor phenotypes by protecting cells from DNA damage [64]. Therefore, the increased expression levels of FTH1 in AOM/DSS mice could have contributed to the development of colon cancer. In contrast, Richard et al. suggested that the overexpression of TfR1 in the tumor tissue compared to that in the normal colon mucosa led to an increased iron content in CRC tumor tissue and increased intracellular LIP, contributing to tumor cell proliferation [65]. Brooks et al. reported that an increase in the expression of iron importer DMT1 and a decrease in iron exporter FPN expression in CRC colon tissue might cause an elevation of cellular iron content and tumor proliferation [66]. However, we did not observe clear changes in the mRNA levels of *TfR1*, *DMT1*, or *FPN* in mice treated with AOM/DSS. Prutki et al. reported that iron metabolism in colorectal cancer could vary depending on the differentiation and metastasis of cancer [67]. They reported that low expression of TfR1 in colorectal cancer tumors was associated with the presence of nodal or distant metastasis in patients [67].

Iron deficiency or iron overload further altered the iron metabolism induced by AOM/DSS. Iron deficiency or iron overload status prior to AOM/DSS treatment completely eliminated the elevated expression of FTH1 in the colon tissues of mice, regardless of iron status. Low FTH1 protein levels in the IOL were not expected because, in iron overload status, the decomposition of ferritin was suppressed [68], but Simao et al. reported that iron overload did not affect the gene expression of *FTH1* in the bone tissue of mice [69]. Low FTH1 levels in either high- or low-iron diet-fed mice could have led to an increase in LIP levels, oxidative stress, and cell death, resulting in the alleviation of colon cancer development in both ID and IOL mice. In contrast, other iron regulatory genes were differentially expressed among the different, iron-diet-fed mice. As expected, TfR1 was upregulated in ID mice, but downregulated under IOL conditions in AOM/DSS mice. The upregulation of TfR1 might have led to increased LIP levels and oxidative stress and inhibited the growth of colon cancer in an ID AOM/DSS model. Simultaneously, the elevation of TfR1 level may indicate that systemic iron deficiency was induced by ID diet, although the ID diet could not further lower the AOM/DSS-induced decrease in iron levels. Cao et al. also reported that iron depletion inhibits the growth of CRC cells [70].

The iron overloaded AOM/DSS model showed an upregulated expression of hepcidin in the liver and DMT1, FPN in the colon and the downregulation of FTH1 in the colon were detected, along with a significant increase in the iron content of the liver and colon when compared to all the other groups. High iron levels have been reported to increase the mRNA levels of hepcidin and colon FPN [71,72,73]. Although we could not measure the protein levels of FPN, high hepcidin levels in the IOL-induced iron overload condition might have lowered the protein levels of FPN, regardless of *FPN* mRNA levels. McDonald’s study also showed that a high-iron diet elevated the transcription of hepcidin and FPN in the livers of mice, but the protein level of FPN did not increase [74]. In the case of DMT1, contrary to our results, it was reported that an iron overload diet lowered the mRNA level of *DMT1* [75]. Therefore, further research is needed to explain these conflicting results, but the increase in the expression of iron importer DMT1 was reported to result in ROS-mediated cell damage [76,77]. The upregulated expression of DMT1 and hepcidin, along with reduced FTH1 expression by iron overload in the AOM/DSS model, might increase the levels of intracellular LIP and ROS-mediated cell damage.

The alteration in the iron state alleviated the aberrant mucosal crypt foci and adenomas seen in AOM/DSS. The number of colonic tumors and tumor rate were significantly reduced in the ID state or modestly reduced in the iron overload state. These changes were accompanied by the downregulated expression of PCNA, cyclin D1, and c-Myc genes, which are key regulators of cancer cells [78,79,80]. PCNA and cyclin D1 have been suggested as hub genes related to CRC development and are powerful therapeutic targets reported to be upregulated in CRC [81]. Previously, iron depletion was reported to induce cell cycle arrest by reducing cyclin D1 levels in mantle cell lymphoma cells [82] and by reducing PCNA and cyclin D1 levels in a dose-dependent manner in immortalized Kaposi’s sarcoma cells [83]. Although chronic iron overload is known to increase cell proliferation [84,85], excessive iron can lead to increased ROS [86] and increased ROS can lead to lipid peroxidation, leading to ferroptosis [87]. Ferroptosis is a type of cell death that can be used as a cancer treatment strategy [88,89]. However, we could not measure the degree of lipid peroxidation in these tissues. In addition, the expression of the *GPX4* gene, which plays a role in preventing ferroptosis, increased. Therefore, we do not know whether ferroptosis occurred in IOL-fed AOM/DSS mice. Nevertheless, excessive ROS production because of high iron concentrations can cause DNA damage, cell cycle arrest, and cell death [90]. We also noticed that the Bax/Bcl-2 ratio was exponentially increased by IOL and slightly increased by ID, suggesting that iron imbalances might have increased cell death and may be involved in reducing tumor severity. It is plausible that decreases in PCNA, cyclin D1, and c-Myc levels with the increase in Bax/Bcl-2 ratio by ID or IOL diet might be associated with the prevention of CRC development.

In addition to the common mechanism of action of the anti-colon cancer effects of iron deficiency or iron overload in the AOM/DSS model, there seemed to be iron status-specific effects. The levels of inflammatory cytokines, including NF-κB, IL-6, TNFα, iNOS, COX2, and IL-1β, were decreased only by IOL diet but not by ID diet compared to those in the normal in AOM/DSS models. NF-κB, a key regulator of inflammatory responses, plays a pivotal role in CAC development [91]. Greten et al. reported that inactivation of the NF-κB pathway through the deletion of IκB kinase in mice’s colons significantly reduced the incidence of CAC tumors following AOM/DSS treatment [92]. In addition, the weakening of NF-κB signaling in the CAC model through various anti-inflammatory substances, such as metformin [93], saponin [94] and fluoxetine [95], inhibited the development of colitis and tumors. The target genes of NF-κB, including IL-6 [96], TNFα [97], iNOS [98,99] and COX2 [100], were also implicated in the CAC tumorigenesis. Grivennikov et al. reported that IL-6-knockdown mice showed reduced CAC tumorigenesis following AOM/DSS treatment [96]. TNF-deficient mice had a reduced size and number of CAC tumors in the AOM/DSS model compared to the control group [97]. In addition, increased expression levels of iNOS and COX2 were reported in colon sections of patients with inflammatory and colorectal cancer and were suggested to be an indicator of tumor development and progression [98,99]. The inhibition of COX2 using nonsteroidal anti-inflammatory drugs reduces the formation of colon tumors in patients with CRC [100]. Conversely, increased levels of IL-1β promote tumor proliferation through the activation of NF-κB signaling in colon cancer cells [101]. Lower levels of NF-κB and its target genes of inflammatory cytokines, as found in the iron-overloaded AOM/DSS model, might have contributed to the inhibition of CAC tumor development. However, there was also a report that excess iron exacerbated colitis and CAC in the AOM/DSS model [39]. In their study, iron was supplied to the AOM/DSS model group in the form of carbonyl iron (1%) for six weeks, and 0.1% iron was supplied for a short or long term. In contrast, our study administered ferrous sulfate to the IOL group for 12 weeks. The liver iron concentration in both studies was approximately 1100 to 1300 mg/kg, showing a significantly higher iron state than to that in the control group. Therefore, in the AOM/DSS model, the biggest cause of the contradictory results for excessive dietary iron was the formulation of dietary iron. Constante et al. reported that high ferrous sulfate (500 mg of iron per kg) alleviated the severity of DSS-induced colitis compared with sufficient ferrous sulfate (50 mg of iron per kg) as a control group. However, high ferric ethylenediaminetetraacetic acid (500 mg of iron per kg), another formulation of dietary iron, worsened colitis compared to the control group [102]. Thus, there might be different responses to high concentrations of ferrous sulfate in colitis, as observed in our study. Upon incorporating 2000 ppm iron sulfate in the diet of CAC mice, indicators of colitis severity, such as the colon length or body weight of the mice, showed no significant differences from the observations in the mice fed with the control diet. However, the tumor burden index and the inflammatory cytokine levels were significantly reduced. Iron overload induced by ferrous sulfate might inhibit the development of CAC by lowering the levels of inflammatory cytokines, and iron overload induced by the administration of the diet containing iron sulfate at a concentration slightly below 2000 ppm is also expected to reduce the severity of colitis. Furthermore, as observed in a previous study [39], diets containing 1% carbonyl iron may rather worsen inflammation and tumor progression. Meanwhile, no significant changes in NF-κB, IL-6, TNFα, and iNOS levels, but higher levels of inflammatory cytokines, such as COX2 and IL-1β, were observed in ID AOM/DSS mice.

In contrast, the activation of the antioxidative pathway has been suggested to mitigate the NF-κB-mediated inflammatory process [103]. The decrease in the expression of the biological antioxidant superoxide dismutase 1 (SOD1) exacerbated DSS-induced colitis in mice through increased ROS-induced oxidative stress and pro-inflammatory immune responses; conversely, an increase in SOD1 expression protected against DSS-induced colitis [104]. Treatment with thioredoxin, a cellular protein oxidoreductase synthesized by *TXN* gene expression, in the human cervical carcinoma cell line HeLa inhibited intracellular NF-κB pathway activation [105]. Furthermore, mice with *GPX* knockout had colitis [106]. Our IOL group showed higher expression levels of genes, including *SOD1*, *TXN*, *GPX1*, *GPX4*, *CAT*, *HMOX1*, and *NQO1* associated with antioxidant pathways, than those seen in all the other groups. Similarly, Moon et al. reported that a dietary iron overload mouse model increased the expression of *TXN*, *HMOX*, and *CAT* in the liver in response to oxidative stress [107]. In particular, the increased activity of *GPX4* inhibits TNFα-induced ROS production and the activation of NF-κB signaling in HEK293T cells [108]. The overexpression of *HMOX* and *NQO1* suppresses lipopolysaccharide-induced expression of TNFα and IL-1β [109]. The enhanced expression of antioxidant genes in IOL-fed AOM/DSS might have contributed to the anti-CAC effects partly by diminishing the inflammatory response.

The alleviation of colon cancer development seen in the iron overload and deficient AOM/DSS models might have been because of a decrease in the AOM/DSS-induced activation of the PI3K/AKT pathway. The PI3K/AKT pathway is overactivated in colon cancer. It also promotes the development of various cancers [110,111,112,113]. The PI3K/AKT pathway promotes cancer growth via cell growth and differentiation, angiogenesis, and cell suicide avoidance signals [114]. In addition, the PI3K/AKT pathway is the most frequently affected pathway in the AOM/DSS mouse model [115]. The overactivation of the PI3K/AKT pathway leads to the overactivation of AKT, and activated AKT promotes cell growth and proliferation [116]. Therefore, studies targeting AKT for cancer treatment continue, and AKT inhibitors that inhibit AKT kinase activity or pAKT expression attenuate the growth of cancer cells [117,118,119]. In addition, AKT inhibitors can induce apoptosis in CRC cells and inhibit tumor growth [114,120,121]. Our data showed that both high- and low-iron diets with AOM/DSS treatment decreased pAKT levels. It has been reported that iron overload lowers the level of pAKT in various cells, leading to cell damage and apoptosis. Ke and Yao reported that iron overload lowered intracellular pAKT and cell viability in preosteoblast cells [122] and bone marrow mesenchymal stem cells [123], respectively. Salama suggested that iron overload downregulated pAKT levels in the livers of mice and caused liver damage [124,125]. In contrast, a decrease in pAKT levels under ID conditions might be associated with increased *Ndrg1* expression. Dixon et al. reported that iron chelator treatment induced tumor suppression by upregulating the metastasis suppressor NDRG1 in prostate cancer cells, and the overexpression of *Ndrg1* induced an increase in the level of the tumor suppressor gene *PTEN* and decreased tumorigenic pAKT [126]. In addition, Kovacevic et al. reported that iron depletion in pancreatic cancer cells upregulated the expression of *Ndrg1*, and the knockdown of *Ndrg1* led to a decrease in the levels of PTEN and pAKT, key molecular targets of PTEN [127]. Thus, our finding of a decrease in pAKT levels under iron-deficient conditions may be associated with an elevation in *Ndrg1* expression and a subsequent increase in PTEN levels.

## 5. Conclusions

Overall, this study suggests that AOM/DSS-induced colon cancer did not induce iron accumulation in the liver and colon of AOM/DSS-treated mice, but rather reduced tissue iron levels, especially LIPs in colon tissue, which might have decreased because of the increase in colonic FTH1. The ID diet led to an increase in TfR1 levels in the colon and a decrease in FTH1 levels. However, iron deficiency might have been maintained to inhibit tumor cell growth. In contrast, the IOL diet might have led to increased levels of LIP in tissues because of upregulated levels of hepcidin and DMT1, and weakened defenses against increased oxidative stress with decreased FTH1 levels, leading to tumor cell death. In addition, the ID and IOL diets alleviated morphological changes in CAC, which might be associated with a decrease in cell proliferation and an increase in apoptosis. The reduced tumor ratio exhibited by the excessive iron diet may account for the restrained expression of inflammatory cytokines and activation of antioxidant-related genes. In addition, it is expected that the reduction in pAKT levels because of excess iron is related to tumor ratio. Paradoxically, despite the decreased colon length and body weight and slightly increased inflammation, AOM/DSS-treated mice fed ID showed a decreased number of tumors, probably because of an increase in *Ndrg1* and PTEN levels owing to iron depletion and the subsequent inhibition of the PI3K/AKT pathway. Overall, we present novel findings that may help to understand the AOM/DSS-treated CAC mouse model, and thereby propose a potent role of iron-overload or -deficient status through dietary iron in CAC treatment.

## Figures and Tables

**Figure 1 nutrients-14-02033-f001:**
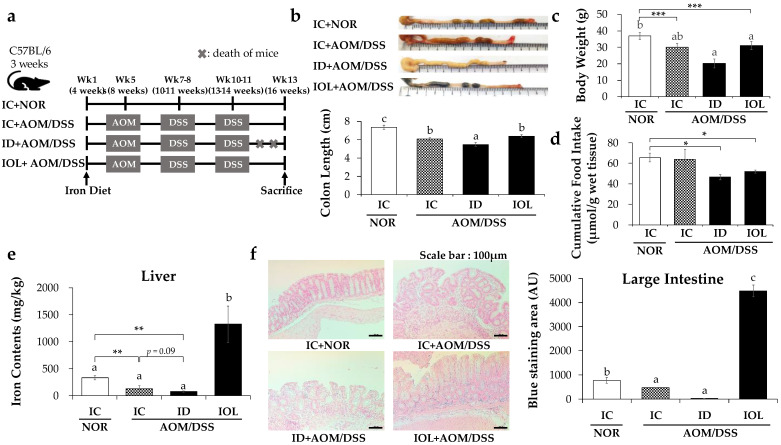
Amount of dietary iron changes inflammation and iron status in the large intestine. Experimental timeline (**a**), and colon length data (**b**) and body weight at final week (**c**), cumulative food intake (**d**) of each group (IC + NOR, IC + AOM/DSS, ID + AOM/DSS, IOL + AOM/DSS). ICP-OES & ICP-MS data measuring the amount of iron in the liver (**e**) and Perls’ Prussian blue staining of the large intestine of mice and the intensity of staining visible within a tissue section (**f**) are presented. Data are expressed as Mean ± SD. One-way ANOVA was used for the test of difference and Tukey for the post-hoc test. Different letters above the value indicate statistical significance. Student’s *t*-test was used to test the difference between two groups, and the asterisk represents statistical significance (* *p* < 0.05; ** *p* < 0.01, *** *p* < 0.001). Scale bar = 100 μm. Abbreviations: NOR, normal; AOM/DSS, azoxymethane/dextran sodium sulfate; IC, iron control; ID, iron-deficient; IOL, iron overload; AU, arbitrary units.

**Figure 2 nutrients-14-02033-f002:**
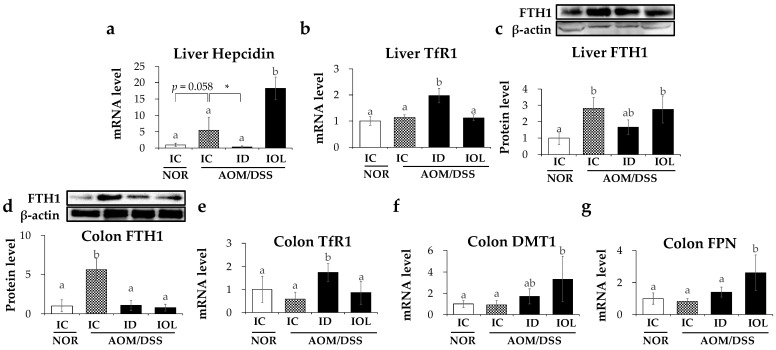
AOM/DSS treatment and iron overload and iron deficient diet affected the expression of iron-related genes in the AOM/DSS model. The hepatic mRNA expression levels of hepcidin (**a**) and *TfR1* (**b**) were evaluated. Western blot analysis of liver FTH1 (**c**) and colonic FTH1 (**d**) was performed. The mRNA expression levels of *TfR1* (**e**), *DMT1* (**f**), and *FPN* (**g**) released from the large intestine tissue were measured. Mean ± SD and SE are represented. One-way ANOVA was used for the test of difference and Tukey for the post-hoc test. Different letters above the value indicate statistical significance. Student’s *t*-test was used to test the difference between two groups, and the asterisk represents statistical significance (* *p* < 0.05). Abbreviations: NOR, normal; AOM/DSS, azoxymethane/dextran sodium sulfate; IC, iron control; ID, iron-deficient; IOL, iron overload; TfR1, transferrin receptor 1; FTH1, ferritin heavy chain 1; DMT1, divalent metal transporter 1; FPN, ferroportin.

**Figure 3 nutrients-14-02033-f003:**
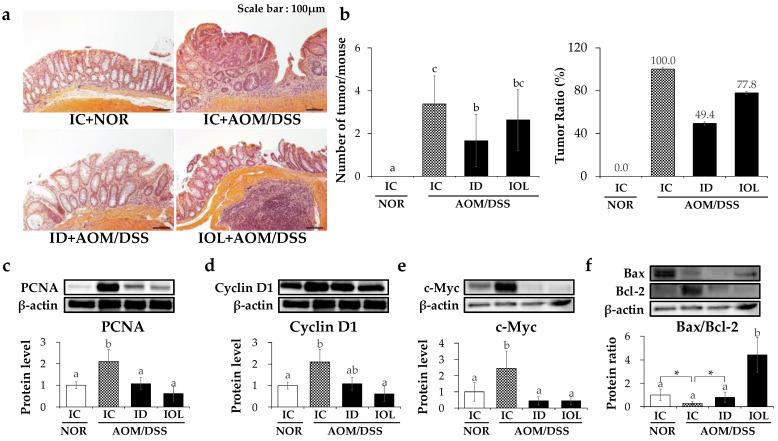
Iron overload or deficiency status suppresses tumor development. Representative images of H&E staining in the large intestine in mice (microscopic view) (**a**). Average tumor number in each group and the proportion of tumors observed in the colons of mice treated with AOM/DSS (**b**). Protein expression levels on PCNA (**c**), Cyclin D1 (**d**), and c-Myc (**e**) and the protein ratio of Bax/Bcl-2 (**f**) released from large intestine tissue. The data are presented as mean ± SD. One-way ANOVA was used for the test of difference and Tukey for the post-hoc test. Different letters above the value indicate statistical significance. Student’s *t*-test was used to test the difference between two groups, and the asterisk represents statistical significance (* *p* < 0.05). Abbreviations: NOR, normal; AOM/DSS, azoxymethane/dextran sodium sulfate; IC, iron control; ID, iron-deficient; IOL, iron overload; PCNA, proliferating cell nuclear antigen; Bax, Bcl-2 associated x; Bcl-2, B-cell lymphoma 2.

**Figure 4 nutrients-14-02033-f004:**
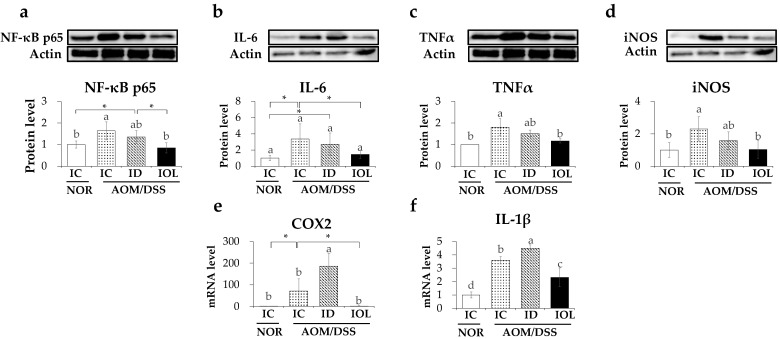
Iron overload downregulated the expression of genes related to AOM/DSS-induced inflammation. Protein expression levels on NF-κB p65 (**a**), IL-6 (**b**), TNFα (**c**), iNOS (**d**), and mRNA expression level of *COX2* (**e**), *IL1b* (**f**) related to inflammation released from large intestine tissue were measured. Data are expressed as Mean ± SD and SE. One-way ANOVA was used for the test of difference and Tukey for the post-hoc test. Different letters above the value indicate statistical significance. Student’s *t*-test was used to test the difference between two groups and the asterisk represents statistical significance (* *p* < 0.05). Abbreviations: NOR, normal; AOM/DSS, azoxymethane/dextran sodium sulfate; IC, iron control; ID, iron-deficient; IOL, iron overload; NF-κB, nuclear factor kappa-light-chain-enhancer of activated B cells; IL-6, interleukin 6; TNFα, tumor necrosis factor alpha; iNOS, inducible nitric oxide synthase; COX2, cyclooxygenase-2; IL-1β, interleukin 1 beta.

**Figure 5 nutrients-14-02033-f005:**
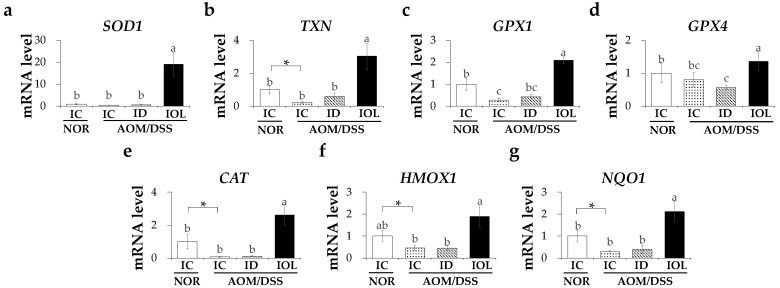
Iron overload diet enhanced the expression of antioxidant pathway genes. mRNA expression level of *SOD1* (**a**), *TXN* (**b**), *GPX1* (**c**), *GPX4* (**d**), *CAT* (**e**), *HMOX1* (**f**) and *NQO1* (**g**) released from the large intestine tissue were measured. Data are expressed as Mean ± SD and SE. One-way ANOVA was used for the test of difference and Tukey for the post-hoc test. Different letters above the value indicate statistical significance. Student’s *t*-test was used to test the difference between two groups and the asterisk represents statistical significance (* *p* < 0.05). Abbreviations: NOR, normal; AOM/DSS, azoxymethane/dextran sodium sulfate; IC, iron control; ID, iron-deficient; IOL, iron overload; *SOD1*, superoxide dismutase 1; *TXN*, thioredoxin; *GPX1*, glutathione peroxidase 1; *GPX4*, glutathione peroxidase 4; *CAT*, catalase; *HMOX1*, heme oxygenase 1; *NQO1*, NAD(P)H quinone dehydrogenase 1.

**Figure 6 nutrients-14-02033-f006:**
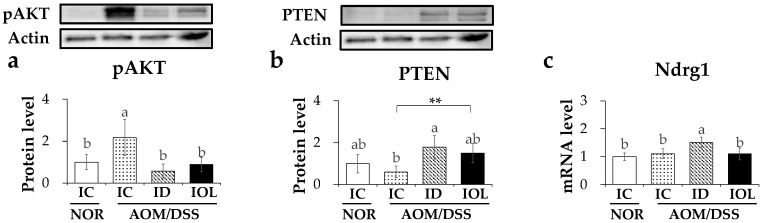
Iron-deficient diet inhibited the activation of the PI3K/AKT pathway, inhibiting the development of CAC. Protein expression level of pAKT (**a**), PTEN (**b**) and mRNA expression level of *Ndrg1* (**c**) released from large intestine tissue were measured. Data are expressed as Mean ± SD and SE. One-way ANOVA was used for the test of difference and Tukey for the post-hoc test. Different letters above the value indicate statistical significance. Student’s *t*-test was used to test the difference between two groups and the asterisk represents statistical significance (** *p* < 0.01). Abbreviations: NOR; normal, AOM/DSS; azoxymethane/dextran sodium sulfate, IC; iron control, ID; iron-deficient, IOL; iron overload. pAKT; phosphorylated protein kinase B, PTEN; phosphatase and tensin homolog, Ndrg1; N-Myc downstream regulated 1.

**Table 1 nutrients-14-02033-t001:** Composition of different iron diets.

Ingredient	AIN-76A Diet Base		
(g/1000 g Diet)	IC	ID	IOL
Casein	200	200	200
Corn starch	150	150	150
Sucrose	499.99	499.99	490.09
Corn oil	50	50	50
Cellulose	50	50	50
Vitamin mixture	10	10	10
AIN 76a mineral mix	35	-	35
AIN 76a mineral mix	-	35	-
(Fe-deficient)			
Choline bitartrate	2	2	2
DL-Methionine	3	3	3
Butylated hydroxytoluene	0.01	0.01	0.01
FeSO_4_, 7H_2_O	-	-	9.9
Total(g)	1000	1000	1000

IC: iron control; ID: iron deficient; IOL: iron overload.

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
