# Peer review of "Supplementation with High or Low Iron Reduces Colitis Severity in an AOM/DSS Mouse Model"

_nutrients, 2022, doi:10.3390/nu14102033_

Round 1

Reviewer 1 Report

In the manuscript "Supplementation with High or Low Iron Reduces Colitis Severity in an AOM/DSS Mouse Model " authors are aimed at evaluating the effects of high or low iron on the colitis severity. Based on reported findings, authors conclude that MccJ25 or reuterine treatments only induce subtle changes of both low and high dose iron could suppress AOM/DSS-induced tumor. Although most of the reported results and discussion are sustained by data, and the workload is quite large, the mechanism is not clear. Some points need to be further clarified as reported in the following points below.

Comments for the authors:

  1. Line 12-14: The description of the experimental design is vague and should be rewritten.
  2. A weight loss of more than 20% is considered a successful modeling. However, there is no data to prove that the model was successfully established. Supplementary experiments are recommended to demonstrate the same level of disease severity baseline in each group.
  3. Fig 4 B and 5 B, Columns with the same letter are still marked with asterisks. Why?

It is recommended to modify the markup of all images. Press from large to small: a, b, c…

  1. A low dose of iron shows a certain relieving effect, and the dose of nearly a hundred times is still non-toxic. Where is the inflection point? It is suggested to supplement the experimental group with higher dose
  2. Line 12-13: Abbreviations should be defined when they are used for the first time. For example, in the sentence “NOR and IC+AOM/DSS, ID+AOM/DSS, and IOL+AOM/DSS, and were fed iron control (IC), iron-deficient (ID)…”, “IC” and “ID” should be defined in the Line 12 instead of in Line 13. Other abbreviations should be checked throughout the manuscript.
  3. Table 1: “1000 g” should be corrected as “1,000 g” or “kg”.
  4. Scale bar should be re-marked with higher resolution in the tissue histology analysis-related figures.

Reviewer 2 Report

The study presented by Moon et al. has investigated the effect of dietary induced excess or deficient iron on tumor progression. Using AOM/DSS-induced colorectal cancer (CAC) rodent model, they studied iron content, gene expression, morphological and histological analysis. The authors showed the comparison of iron metabolism and status in AOM/DSS-treated mice by controlling iron overload or deficiency. In addition, the progression of colonic tumor was alleviated in IOL and ID diets. Although many studies have comprehensively dedicated to the effects of excess or deficient of iron, metabolism is still complicated. However, Moon et al, the results of this study is well demonstrated, well-written, the data are solid, and the findings are of interest to the readership of this journal. 

  1. Line 387, Compared to Kim et al. about iron diet administration (6 weeks), why you used longer administration (13 weeks)? It seemed that the content of iron in the model might affect colonic expression of FTH1, cytoplasmic LIP level, or else. After all, it also can change of significance in result. Any standard (basis) on this diet duration?
  2. acronym; please show full-term these words, if it is correct you assumed.

Line 55, ulcerative colitis (UC)

Line 62, Crohn's disease (CD)

Line 110, NOR (Normal?)

Round 2

Reviewer 1 Report

The authors adressed the questions well.